# Multi-laboratory assessment of reproducibility, qualitative and quantitative performance of SWATH-mass spectrometry

Ben C. Collins [1], Christie L. Hunter[2], Yansheng Liu [1], Birgit Schilling[3], George Rosenberger [1,4],
Samuel L. Bader[5], Daniel W. Chan[6], Bradford W. Gibson[3,7], Anne-Claude Gingras[8,9], Jason M. Held[10],
Mio Hirayama-Kurogi[11], Guixue Hou[12], Christoph Krisp[13], Brett Larsen[8], Liang Lin[12], Siqi Liu[12], Mark P. Molloy[13],
Robert L. Moritz [5], Sumio Ohtsuki[11], Ralph Schlapbach[14], Nathalie Selevsek[14], Stefani N. Thomas [6],
Shin-Cheng Tzeng[10], Hui Zhang[6] & Ruedi Aebersold[1,15]

Quantitative proteomics employing mass spectrometry is an indispensable tool in life science research. Targeted proteomics has emerged as a powerful approach for reproducible quantification but is limited in the number of proteins quantified. SWATH-mass spectrometry consists of data-independent acquisition and a targeted data analysis strategy that aims to maintain the favorable quantitative characteristics (accuracy, sensitivity, and selectivity) of targeted proteomics at large scale. While previous SWATH-mass spectrometry studies have shown high intra-lab reproducibility, this has not been evaluated between labs. In this multi-laboratory evaluation study including 11 sites worldwide, we demonstrate that using SWATH-mass spectrometry data acquisition we can consistently detect and reproducibly quantify >4000 proteins from HEK293 cells. Using synthetic peptide dilution series, we show that the sensitivity, dynamic range and reproducibility established with SWATH-mass spectrometry are uniformly achieved. This study demonstrates that the acquisition of reproducible quantitative proteomics data by multiple labs is achievable, and broadly serves to increase confidence in SWATH-mass spectrometry data acquisition as a reproducible method for large-scale protein quantification.

[1] Department of Biology, Institute of Molecular Systems Biology, ETH Zurich, 8093 Zurich, Switzerland. [2] SCIEX, 1201 Radio Road, Redwood City, CA 94065, USA. [3] Buck Institute for Research on Aging, 8001 Redwood Boulevard, Novato, CA 94945, USA. [4] PhD. Program in Systems Biology, University of Zurich and ETH Zurich, Zurich 8057, Switzerland. [5] Institute for Systems Biology, 401 Terry Avenue North, Seattle, WA 98109, USA. [6] Department of Pathology, Clinical Chemistry Division, Johns Hopkins University School of Medicine, Baltimore, MD 21231, USA. [7] Department of Pharmaceutical Chemistry, University of California, San Francisco, CA 94143, USA. [8] Lunenfeld-Tanenbaum Research Institute, Sinai Health System, Toronto, M5G 1X5 Ontario, Canada. [9] Department of Molecular Genetics, University of Toronto, Toronto, M5S 1A8 Ontario, Canada. [10] Departments of Medicine and Anesthesiology, Washington University School of Medicine, 660 South Euclid Avenue, St. Louis, MO 63110, USA. [11] Department of Pharmaceutical Microbiology, Faculty of Life Sciences, Kumamoto University, 5-1 Oe-honmachi, Chuo-ku, Kumamoto 862-0973, Japan. [12] Proteomics Division, BGI-Shenzhen, Shenzhen 518083, China. [13] Department of Chemistry and Biomolecular Sciences, Australian Proteome Analysis Facility (APAF), Macquarie University, Sydney, 2109, Australia. [14] Functional Genomics Center Zurich, ETH Zurich/University of Zurich, Winterthurerstr. 190, 8057 Zurich, Switzerland. [15] Faculty of Science, University of Zurich, Zurich, Switzerland. Ben C. Collins, Christie L. Hunter, and Yansheng Liu contributed equally to this work. Correspondence and requests for materials should be addressed to R.A. (email: aebersold@imsb.biol.ethz.ch)

Reproducibility is an essential foundation of scientific research. Recent reports have concluded that a significant fraction of life science research shows poor reproducibility of results and this poses a major challenge to scientists, science policy makers, funding agencies, and the pharma and biotech industry sectors[1–3]. The reasons for irreproducibility of research results are many, including inadequate study design and data analysis, limited data quality, incompletely characterized research reagents, poorly benchmarked techniques, and a range of other confounding factors.

The question of whether specific data acquisition methods and platforms are capable of generating reproducible results is best addressed by inter-laboratory studies, where samples of known composition and quality are analyzed across different settings. Such studies have been reported for various "omics" technologies, including RNA-seq and microarray techniques, with varying results[4, 5]. Such projects have served to highlight problems in various large-scale strategies, to stimulate discussion in a given field on how to improve reproducibility, and in the best cases to provide confidence in a given strategy within and beyond an analytical field.

In the field of mass spectrometry (MS) based proteomics, a wide range of specific methods have been reported over the past two decades. These can be broadly grouped into discovery and targeted proteomic techniques. The general aim of discovery proteomics is the unbiased identification and quantification of the protein components of biological samples. This is most frequently achieved by data-dependent acquisition (DDA) MS. If the number of precursor ions exceeds the number of precursor selection cycles[6], precursor selection becomes stochastic and the peptides detected in repeat analyses become irreproducible. This has been documented in a number of intra- and inter-laboratory studies[7–9]. In general, these studies confirmed that a high degree of reproducibility is difficult to achieve for complex samples[10]. Computational methods to enable improved quantification via propagation of peptide identifications across runs via alignment of MS1 precursor signals, first introduced as accurate mass and time tags (AMT)[11, 12], are commonly applied to DDA data[13–16] and can reduce this issue to some degree in discrete data sets where chromatographic alignment can reasonably be applied.

In contrast to discovery proteomics the general aim of targeted proteomics is the detection and quantification of a predetermined set of peptides by selected reaction monitoring (SRM) also known as multiple reaction monitoring (MRM)[17], or a related technique parallel reaction monitoring[18–20]. Because targeted MS eliminates the stochastic component of precursor ion selection in DDA, it has the potential for high reproducibility. This has been demonstrated in intra-laboratory studies where sets of peptides were targeted with a high degree of reproducibility across relatively large sample sets[21–23] and by inter-laboratory studies focused on exploring the use of SRM and immuno-SRM for biomarker studies[24–29]. Targeted MS is now broadly regarded as a reproducible protein analysis platform[17]. However, the number of proteins measured is restricted (usually to ~100 per injection), limiting its utility for many applications.

SWATH-MS is a more recently introduced approach to MS-based proteomics[30]. It consists of data-independent acquisition[31, 32] (DIA) in which all precursor ions within a user defined $m/z$ window are deterministically fragmented. Analysis of SWATH-MS data most often relies on a targeted data analysis strategy in which target peptides are detected and quantified from the SWATH-MS fragmentation data by extracting and correlating previously generated query parameters for each target. In this scheme each unique peptide of interest at a given precursor charge state is queried for in the data, resulting in the detection and scoring of co-eluting transition group signals and associated underlying mass spectral features, referred to as peak groups. Because the method specifically tests for the presence of each target peptide in the essentially complete fragment ion map of each sample, it eliminates the stochastic sampling element of DDA and helpfully provides a direct statistical measure (e.g., $q$-value) of whether the peptide is present at a detectable level in the sample. This data analysis strategy, whereby target peptides are directly queried for, has recently been generalized using the term peptide-centric[33, 34] scoring to distinguish from more classical approaches where the MS2 spectrum is the query unit for data analysis (referred to as spectrum-centric scoring). The SWATH-MS implementation of the DIA concept therefore preserves the favorable performance characteristics of SRM, while vastly expanding the measurement capacity to thousands of proteins per injection. Of consideration in SWATH-MS is the complexity of the resultant spectra and specific software tools have been compiled to analyze such highly multiplexed data using various approaches[35–38]. A recent study comparing software tools for the analysis of DIA data using either peptide-centric or spectrum-centric approaches has demonstrated that very similar qualitative and quantitative results can be obtained when analyzing a benchmarking data set[39]. SWATH-MS and related DIA approaches have achieved a high degree of reproducibility in intra-laboratory studies in a variety of research questions such as interaction proteomics[40, 41], plasma proteomics[42], tissue proteomics[43], microbial proteomics[44, 45], pre-clinical toxicology[9], analysis of genetic reference strains[46], and many others. However, interlaboratory robustness and reproducibility of SWATH-MS data acquisition has not been demonstrated.

In this study, we set out to test the reproducibility of peptide and inferred protein detection and quantification by SWATH-MS in an inter-laboratory study. To achieve this goal we distributed benchmarking samples to 11 participating laboratories worldwide for measurement by SWATH-MS according to a predetermined schedule. We analyzed the data from all sites centrally with two separate scopes in mind. Firstly, we analyzed all of the data in an aggregated way to simulate, for example, a large cohort study whereby patient samples would be analyzed in multiple laboratories, aiming to achieve a result set based on all samples. In the second interpretation, we analyzed the data from each site of collection independently and compared the results across sites post analysis facilitating a direct performance comparison.

Our analysis demonstrated that the set of proteins detected and quantified across all participating sites, i.e., from a total of 229 proteome measurements, was very consistent. The reproducibility, linear dynamic range, and sensitivity are approaching those reported for SRM, currently the gold standard approach for protein quantification[17, 30, 47]. This data supports the conclusion that DIA combined with peptide-centric scoring embodied by the SWATH-MS approach is suitable for both comprehensive and reproducible proteomics at a large scale and across laboratories.

## Results

**Study design and implementation.** To assess the inter- and intra-laboratory reproducibility and performance of SWATH-MS for large-scale quantitative proteomics, we created a benchmarking sample set and distributed aliquots to 11 laboratories worldwide (Fig 1a). The sample consisted of 30 stable isotope-labeled standard (SIS) peptides[48] diluted into a complex background consisting of 1 µg of protein digest from HEK293 cells. To achieve both, a physiologically relevant fold change step, and to cover a large linear dynamic range in a relatively small number of samples that could be analyzed in a 24 h period, we elected to partition the SIS peptides into five groups (A–E), each containing six peptides. In each group, the dilution series started from a

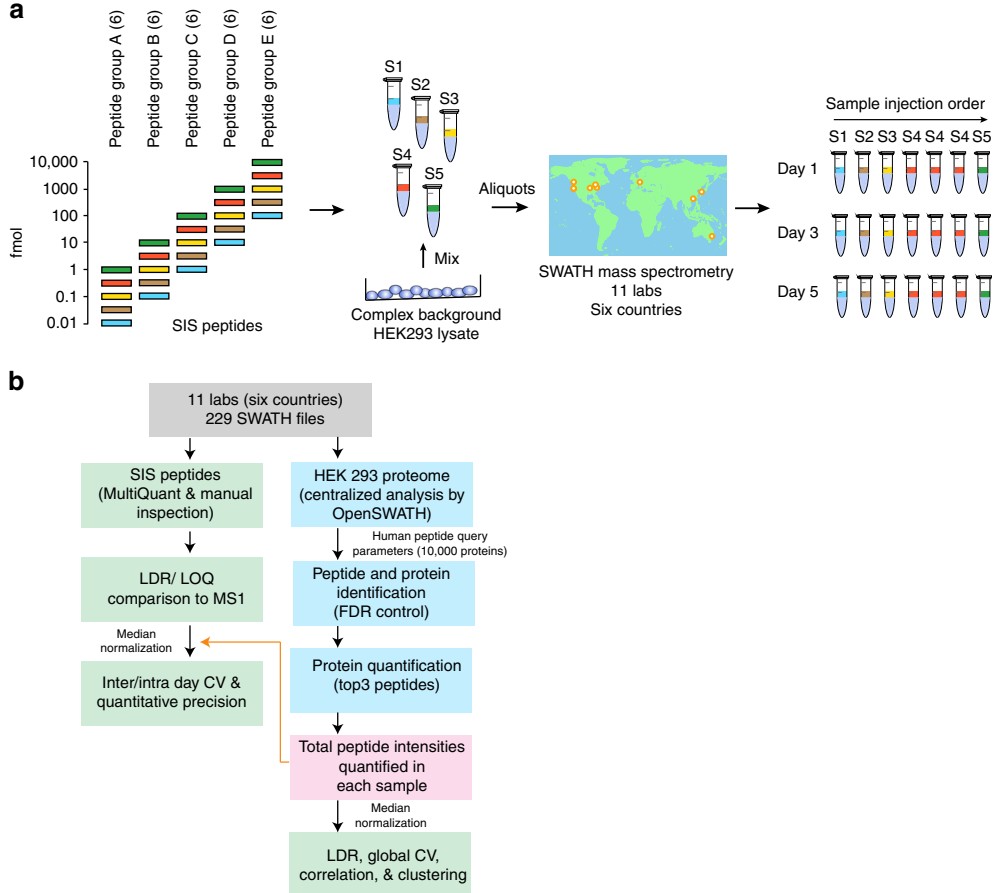

**Fig. 1** Study design and implementation. **a** A set of 30 SIS peptides partitioned into five groups (A–E, six peptides in each) were diluted into a HEK293 cell lysate to span a large dynamic range. Starting at a different upper concentration for each group, they were threefold diluted into the matrix to cover a concentration range from 12 amol to 10 pmol in 1 µg of cell lysate. This created a set of five samples to be run by SWATH-MS on the TripleTOF 5600/5600+ system at each site. Each sample was run once per day on day 1, 3, and 5, with the exception of sample 4 which was run 3× on each day. **b** After data acquisition, the 229 SWATH-MS files were assembled centrally and processed using two strategies. The SIS peptide concentration curves were assessed using MultiQuant Software, allowing for the determination of linear dynamic range (*LDR*), and LLOQs for each peptide. In addition, the intra- and inter-day CVs were determined before and after normalization. The HEK293 proteome matrix data was analyzed using the OpenSWATH pipeline and the Combined Human Assay Library consisting of ~10,000 proteins. The false discovery rate was controlled at the peptide query and protein level using PyProphet. Protein abundances were inferred by summing the top five most abundant fragment ions from the three most abundant peak groups using the aLFQ software. We then used protein abundances to cluster, and compute Pearson correlation coefficients, for all samples from all sites

different level ranging from 1 fmol to 10 pmol (sample S5). The MS responses of the peptides were measured, ranked, and they were assigned evenly to the five groups (A–E) to ensure there was a range of peptide responses across in each group concentration group. The peptides were then diluted serially threefold into the HEK293 background four times (samples S1–S4). This generated an overall dilution series from 0.012 to 10,000 fmol on column, with a linear dynamic range over six orders (although not covered by any single SIS peptide—Supplementary Data 1 and 2). We acquired all data in SWATH-MS mode, set to 64 variable width Q1 windows chosen to minimize window size in high density precursor ion ranges (Supplementary Data 15).

To standardize the SWATH-MS acquisition protocol and to make an initial quality assessment, we first asked each site to acquire five replicate injections of a test sample containing only the HEK293 background. This data was used to improve quality control procedures and to ensure adequate system performance at all sites (Supplementary Fig. 1; Supplementary Note 1). The finalized study protocol is provided (Supplementary Methods). All sites used the same mass spectrometer (SCIEX TripleTOF 5600 / 5600+ systems), while the nanoLCs consisted of various models from the same vendor (SCIEX). The chromatographic

columns had the same dimensions (30 cm × 75 µm) although nine sites used cHiPLC microfluidic systems and two sites used self-packed columns with emitters and, as such, therefore also used different chromatographic resins (see "Methods"). After the initial quality control phase, participating labs acquired SWATH-MS data for the main sample set consisting of samples S1–S5 with sample S4 injected in technical triplicate, and repeated this acquisition scheme two further times during 1 week. The purpose of this design was to determine reproducibility and quantification metrics within 1 day, across 1 week, or across different sites of data collection. These measurements resulted in a data set containing in total 229 SWATH-MS files from the 11 sites worldwide that are freely accessible for further analysis by the community.

**Consistency of protein detection**. The qualitative similarity of SWATH-MS data acquired at different sites was investigated by comparing the set of inferred proteins detected from the HEK293 proteome across all 229 SWATH-MS data files. Targeted analysis was performed using the OpenSWATH software[35] combined with a previously published SWATH-MS spectral library

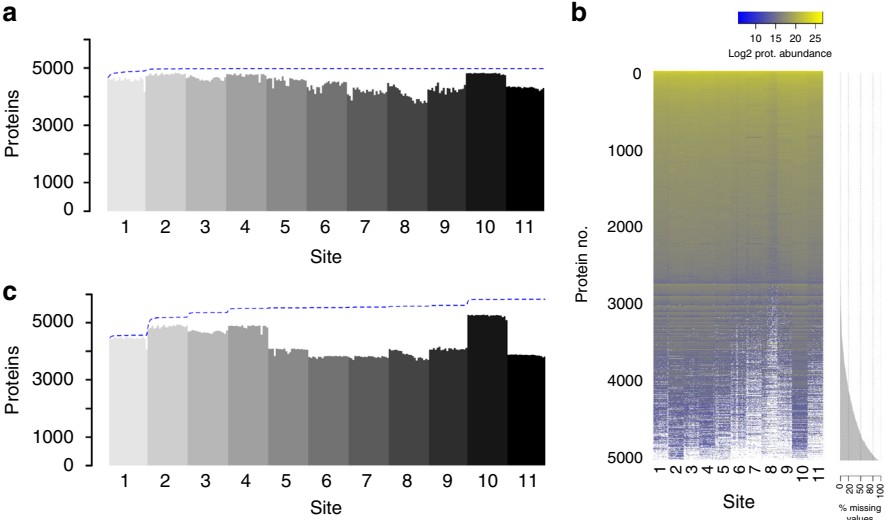

**Fig. 2** A consistent set of proteins is detected across sites. **a** The number of proteins detected in each of the 229 SWATH-MS analyses is shown ordered by site of data collection and then chronologically by time of acquisition. After filtering the data set in a global fashion at 1% FDR at the peptide query and protein levels, a protein was considered detected in a given sample when a peak group for that protein was detected at 1% FDR in the context of that sample (see Supplementary Note 2 for a detailed discussion of FDR). The *blue line* indicates the cumulate set of proteins detected with each new sample moving from left to right. The maximum of the *blue line* indicates the set of proteins detected at 1% FDR in the global context. The saturation of the number of proteins detected after a few samples indicates that the set of proteins observed by all sites is highly uniform. **b** A protein abundance matrix on the log2 scale is shown for 229 SWATH-MS analyses from all sites corresponding to the set of proteins shown in **a**. *White* indicates a missing protein abundance value where a given protein was not confidently detected in a given sample. The proteins are ordered from top to bottom first by row completeness and then by protein abundance. **c** Equivalent to **a** except that the analysis and FDR control is carried independently out on a site-by-site basis instead of aggregated across all sites before analysis and FDR control

containing peptide query parameters mapping to 10,000+ human proteins[49] (Fig. 1b). The false discovery rate (FDR) was controlled at 1% at the peptide query and protein levels using the *q*-value approach[50–54] in the global context, and at 1% peptide query FDR on a sample-by-sample basis. We did not employ any alignment or transfer of peptide identification confidence between runs. A description of FDR calculation, and issues surrounding this, is provided in Supplementary Note 2 and in a related paper explaining FDR considerations in detail[55].

The results are shown in Fig. 2. In Fig. 2a, we depict the number of proteins detected across all SWATH-MS acquisitions in the aggregated data analysis (equivalent plot at peptide query level in Supplementary Fig. 2). The total number of proteins detected at 1% FDR over the entire data set is 4984 from 40,304 proteotypic peptide peak groups (Supplementary Data 3). The median number of proteins detected per file is 4548 from a median of 31,886 peak groups. A total of 4077 proteins were detected in >80% of all samples. Figure 2b shows the distribution of complete/missing values from this data. Of the 4984 proteins detected, 3985 were detected using >1 peptide peak group and, on average, we detected 8.1 proteotypic peptides per protein (Supplementary Fig. 3). Information regarding mass spectrometric and chromatographic performance metrics across the sites that might affect the number of proteins detected is provided in Supplementary Figs. 4–9. The accumulation of new protein identifications over the data set—indicated by the *blue curve* in Fig. 2a—saturates, indicating the comprehensiveness of the SWATH-MS methodology and the minimal number of accumulated false positive identifications across 229 measurements. This also indicates that when we analyzed the data in an aggregated manner (i.e., data from all sites combined), the set of proteins detected by all labs is very consistent. Achieving this consistency was dependent on appropriate FDR control in the global context at both peptide query and protein level. To illustrate this, we plotted the numbers of peak groups and proteins detected when

FDR was controlled only at peptide query level and not the protein level, and only on a sample-by-sample basis and not in the global context (Supplementary Fig. 10). The accumulation of new peak groups steadily increased across the data set, indicating a likely accumulation of false positives and, highlighting the importance of appropriate global FDR control[55, 56]. We computed the repeatability of detection at the peptide and protein levels, similar to Tabb et al.[7], defined as the pairwise percent overlap between any two runs. The range of median repeatability within sites was 90.0–98.2% at the protein level and 79.5–95.5% at the peptide level (Supplementary Fig. 11). The median repeatability over the entire data set from all sites was 91.6% at the protein level and 79.5% at the peptide level.

The comparison between the protein detection rates from the aggregated analysis and an individual site-by-site analysis also provides insight into FDR control. Figure 2c shows the number of proteins detected when the data from each site was first analyzed separately by site of data collection with independent FDR control and then aggregated (equivalent plot at peptide query level in Supplementary Fig. 12). In this analysis, the procedure was identical to that of the aggregated analysis, except that the global context for FDR control mentioned above was restricted to the files from an individual site, and that procedure was repeated for each site individually. The information content of the data from each site is different, which likely relates to performance differences between chromatographic, nanospray ionization and/ or instrument efficiencies across sites at the time of data acquisition. When the data is aggregated before analysis and FDR control, the higher quality data effectively supports the lower quality data, because the strict scoring cutoffs required by the 1% protein FDR threshold only needs to be achieved once per protein in the global context, leading to more homogenous results in terms of proteins detected. That is, in our analysis, a protein is considered detected in a given sample if it is detected at the 1% peptide query FDR threshold as long as the peptide has been

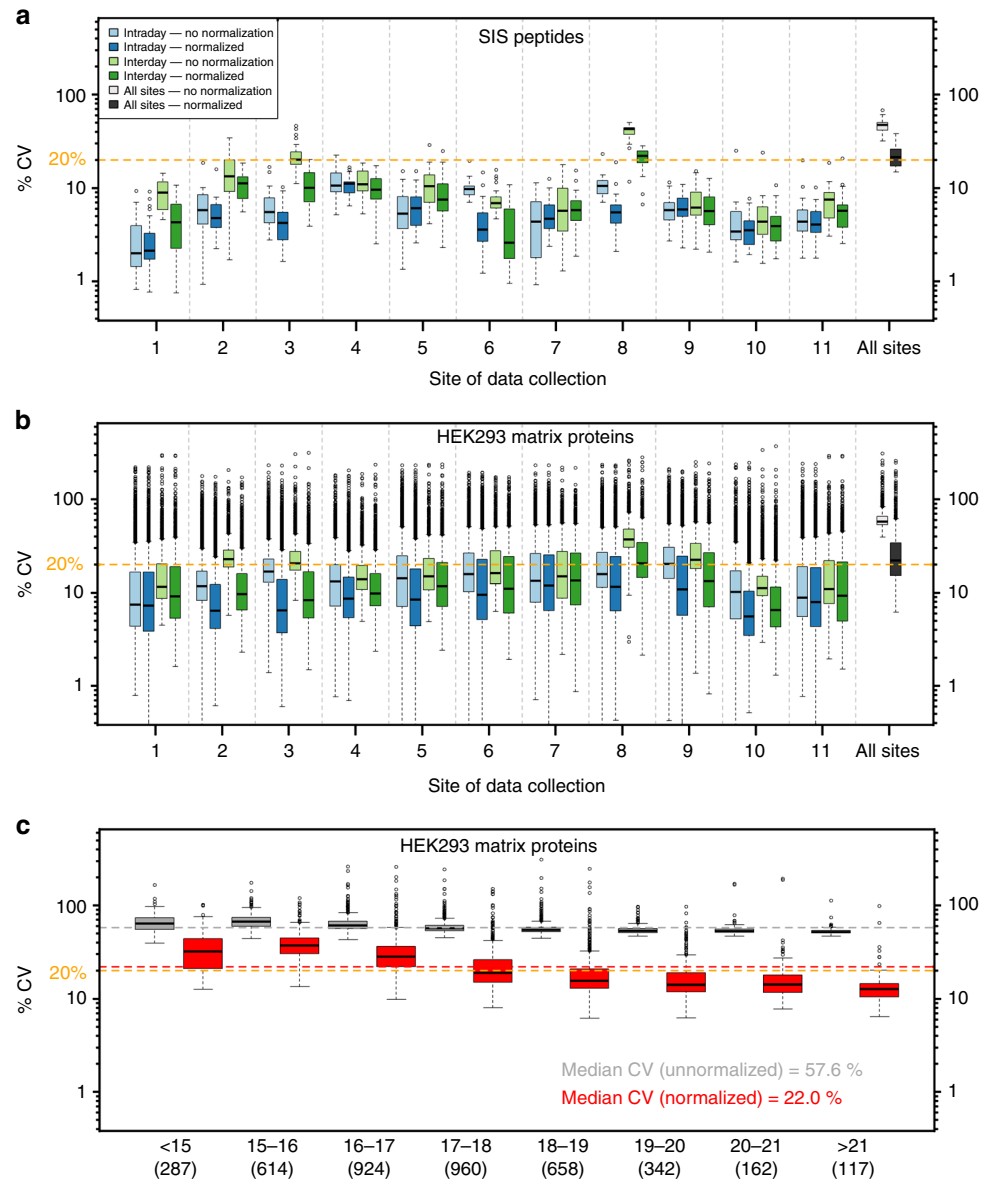

**Fig. 3** Reproducibility of SWATH-MS measurements. **a** The CVs of peak areas for each of the 30 SIS peptides for S4 sample, depicted on the y-axis using logarithmic scaling, were determined at the intra-day level within the site (*light blue*—without normalization, *dark blue* with normalization), inter-day level within site (*light green*—without normalization, *dark green* with normalization), and inter-site level (i.e., over all S4 samples in the study; *light gray*—without normalization, *dark gray*—with normalization). The *orange line* indicates 20% CV for visual reference. **b** Similarly, the CV of protein abundances for the 4077 proteins that were detected in >80% all samples were computed at the intra-day level within the site, inter-day with site, and inter-site (i.e., all 229 samples in the study). **c** The inter-site CVs were binned based on log2 protein abundance to visualize the dependence of CV on protein abundance

detected elsewhere in the experiment with a score passing the 1% protein FDR threshold (Supplementary Note 2).

From these analyses, we can conclude that using SWATH-MS data collected from instruments in different labs, the set of proteins detected is comparable (Fig. 2a, b). This presents a desirable quality not previously demonstrated at this scale in large-scale proteomics analysis.

**Reproducibility of quantification.** Having established a high degree of reproducibility of protein detection within and across sites, we went on to investigate the quantitative characteristics of our inter-lab SWATH-MS data set. To determine quantitative reproducibility we computed the coefficient of variation (CV) at

different levels. Firstly, we extracted ion chromatograms (XIC) for the SIS peptides and summed the XICs to obtain peptide peak areas using the MultiQuant software (Supplementary Data 4). Next, we computed the CV for each site within 1 day (intra-day) and over the week (inter-day) for the S4 sample, which was acquired every day in triplicate. The median for site intra-day and inter-day CVs (expressed as median ± standard deviation) were 5.5 ± 2.9% and 8.9 ± 11.1%, respectively (Fig. 3a, Supplementary Data 4 and 5). For the majority of sites the intra-day and inter-day CVs were below 20% (one lab—Site 8—experienced some larger LC–MS variance over the course of the week with decreasing signals that was later explained by a contaminated collision cell). As the signal response varies between instruments, attempting to directly compare raw peak area or intensities across

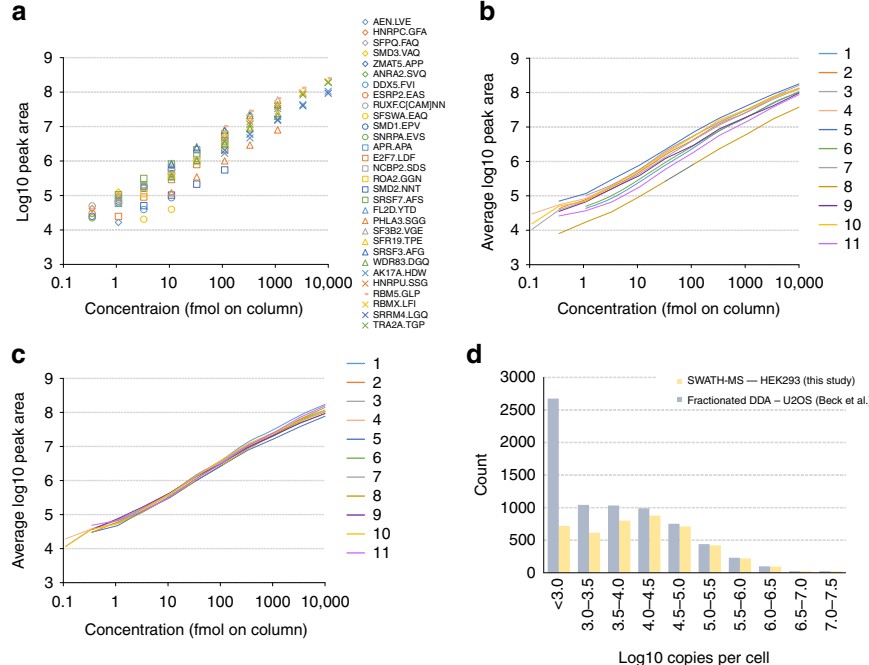

**Fig. 4** Dynamic range and linearity. **a** The response curves for each of the 30 SIS peptides for Site 1 were determined and plotted together (corresponding plots for all other sites are shown in Supplementary Fig. 13). **b** From this data, an average response curve for each site was constructed by averaging (mean) the responses of peptides at the same concentration point. This visualization facilitates comparison of both the dynamic range and average response between sites. **c** The average response curves from **b** replotted after the normalization has been applied. **d** The proteins detected in the SWATH-MS analysis of the HEK293 proteome matrix were mapped onto a previous in-depth DDA analysis of the U2OS cell line that employed multi-level fractionation to achieve deep proteome coverage. To demonstrate the dynamic range achieved by the single-shot SWATH-MS analysis we plotted the proteins detected by SWATH-MS binned by the protein copies per cell value (log10 scale) determined from the in-depth U2OS DDA study[60]. In the range $10^5$–$10^7$ copies per cell the proteome coverage is essentially complete and decreases with lower copies per cell bins

sites is not feasible. To determine if we could normalize the instrument response differences by applying a simple normalization, we used the quantitative information from the HEK293 proteome that is expected to be invariant. Specifically, the peptide peak areas from the automated OpenSWATH analysis for each of the 229 files were re-scaled such that the median values from each file were equalized. The resulting protein abundance boxplots in Supplementary Fig. 13 clearly shows the effect of this simple normalization. The normalization coefficients (Supplementary Fig. 14) were used to adjust the peptide peak areas for the SIS peptides derived from the MultiQuant analysis and the intra-day and inter-day CV analysis was repeated (Fig 3a). We then calculated the inter-site CVs for the SIS peptides using all measurements of the S4 sample from all sites. The median of the inter-site CVs using peptide peak areas without normalization was 47.3 ± 13.9%. After normalization, this was reduced to 21.3 ± 10.3%. Normalization also reduced the median within site inter-day CV from 8.9 ± 11.0 to 5.8 ± 5.4% whereas the intra-day CV was less strongly affected (5.5 ± 2.9 to 4.7 ± 2.3% intra-day CV) (Supplementary Data 4 and 5). The CVs obtained are in a range comparable with previous direct comparisons of SWATH-MS and SRM[47].

We next elected to examine the CV at protein level in the HEK293 proteome across 21 SWATH-MS acquisitions at each site. Protein level abundances were inferred from the OpenSWATH results by summing the top five most intense fragment ion areas from the top three most intense peak groups per protein[42, 44, 57] (Supplementary Data 6 and 7). For proteins where <3 peak groups were detected, all the available fragments were summed. The CVs, computed from the 4077 proteins that were detected in >80% of all samples, at the intra-day, inter-day, and inter-site levels were 8.3 ± 16.2, 11.9 ± 17.2, and 22.0 ± 17.4%

respectively, after peptide level median normalization (Fig. 3b). The inter-site protein CV as a function of protein abundance is shown in Fig 3c.

**Linearity and dynamic range**. To determine the linearity and dynamic range characteristics of SWATH-MS data within and across the sites we first examined the dilution series of SIS peptides in response curves generated from the MultiQuant Software XIC analysis. A representative example for a single site is shown in Fig 4a (remaining sites in Supplementary Fig. 15; equivalent plots separated by peptide are shown in Supplementary Fig. 16; source data in Supplementary Data 8). Peak integration for the lowest concentration peptides was manually inspected to confirm correct peptide detection and that lower limits of quantitation conformed with good bioanalytical standards (<20% CV, 80–120% accuracy, and $S/N > 20$ at the lower limit of quantitation (LLOQ)[58]. Low concentration data points failing these assessments were removed and the next higher concentration was evaluated. This was repeated until a good LLOQ was found. Manual integration adjustments were only done in the cases where there were clear interferences that could be removed.

To obtain an overview of the linearity and dynamic range of the SWATH-MS method between sites, we computed the average peptide peak area (unnormalized) of the SIS peptides at a given concentration point and plotted this as averaged response curve for each site (Fig 4b, Supplementary Data 9). By averaging over six peptides that have variable responses we obtained a representative picture of the linearity and dynamic range of the method (as opposed to that of individual peptides that are more frequently of greater interest in targeted proteomics studies which employ dilution curves). The linear regressions for the average

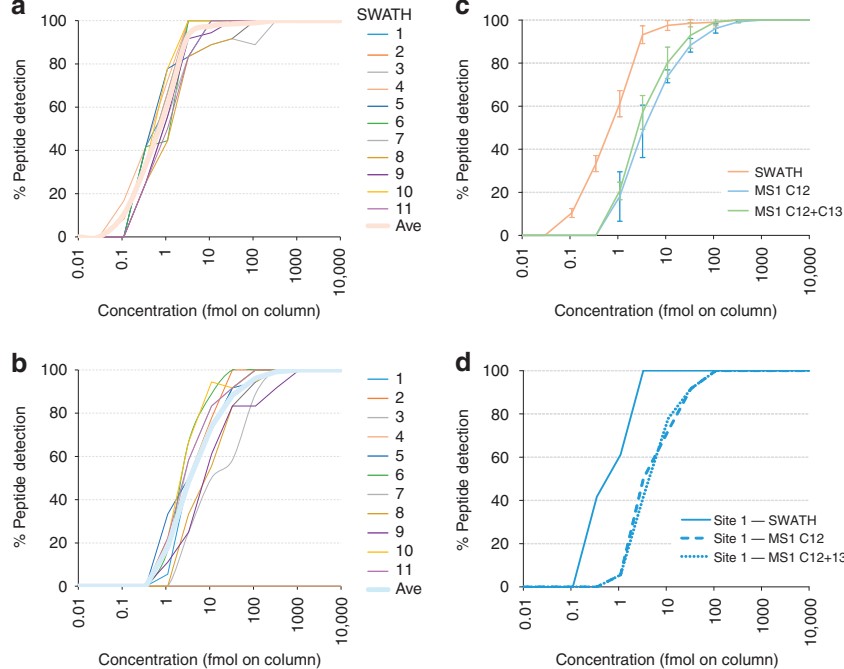

**Fig. 5** Lower limit of quantification in SWATH-MS and MS1. The percentage of the 30 SIS peptides detected at each concentration in the dilution series from each site of data collection was plotted at the SWATH-MS level **a** and the MS1 level **b**. Lower limit of quantification was defined as <20% CV, $S/N >$ 20, 80–120% accuracy using linear fit with $1/x$ weighting in the response curve. Spectral peak widths for XIC generation were 0.02 $m/z$ for MS1 and 0.05 $m/z$ for SWATH-MS2, and the nominal resolving power was 30,000 and 15,000, respectively. **c** The average % detection at each concentration for all sites was determined (*bold line* in **a** and **b**) and overlaid to summarize detection differences between SWATH-MS and MS1 data. For the MS1 data, the C12 and C13 XIC data was also summed for comparison. Error bars are ± 1 standard deviation. **d** The data from a single site (site 1) is also shown for comparison

peptide area curve for each site was computed and the $R^2$ values averaged 0.97 ($R^2$ values for individual peptides are in Supplementary Data 8). There was signal saturation for the highest concentration point (10,000 fmol), and removal of that point increased the $R^2$ to 0.99. Since this study was performed, a newer instrument platform (TripleTOF 6600) has increased linear dynamic range through a different detection system, and signal saturation at high peptide load would be significantly reduced in this case. The average response curves were very similar between sites, all exceeding 4.45 orders of linear dynamic range including all data points, with an average across sites of 4.6 (Supplementary Data 9). Dynamic range was computed by taking the log base 10 of the concentration of the highest point divided by the LLOQ concentration.

By applying the average peak area and average response curves, the data showed that the linearity and dynamic range for each site is qualitatively similar in terms of slope and span. The raw peak areas obtained from each site, however, are offset by a fixed amount across the dynamic range. When the same averaged response curve plot was constructed from values normalized based on the HEK293 proteome background, the response curves were well overlaid (Fig. 4c). The peptide peak area fold change between dilution steps averaged 2.66 across the concentration range, reflecting the three-fold dilution series (ratios in the middle of the linear dynamic range are close to 3 with some compression[59] of the ratio at the lowest and highest concentration points—Supplementary Fig. 17). The mean fold change for expected ratios of ninefold and 27-fold were 7.49 and 19.6, respectively. The ratio compression is partly explained by the high peptide loads (low pmol on column range) used at the upper end of the dilution series, higher than are commonly used for this experiment type, which caused some MS signal saturation.

We next attempted to assess the dynamic range of the measurements at the protein level in the HEK293 proteome. At the protein level, no internal standard was available on which to judge dynamic range. Therefore, as a surrogate measure, we mapped the set of proteins detected in our experiment onto a previous in-depth proteomic characterization of U2OS cells which estimated the copy numbers of proteins per cell[60]. Although the reference data is from a different cell line, an in-depth quantitative comparison of these two cell lines has shown that the protein abundances are well correlated (Pearson correlation ~0.8)[61] making this a reasonable surrogate measure. From this data we can estimate that the set of proteins detected by SWATH-MS in the HEK293 cell proteome spans ~4.5 orders of magnitude, with the upper ~2.5 orders of magnitude being highly complete (Fig. 4d).

**Sensitivity in SWATH-MS and MS1**. Based on the experimental design, there is an expected number of the SIS peptides that could be detected at each concentration (Fig. 1a). To get a broad view of the LLOQ across the study, we plotted the percentage of the 30 SIS peptides that were reliably detected (LLOQ and above) at each concentration in the dilution series from each site (Fig. 5a, Supplementary Data 10). Interestingly, the curves depicting % detection of peptides for the SWATH-MS data across different sites of data collection are uniform, indicating that consistent sensitivity can be achieved at different sites despite the high complexity background. The LLOQ for SWATH-MS data spanned the mid-attomole to low-femtomole range. Despite the higher complexity background proteome used in this study, the results are in good agreement with data previously obtained[30, 47].

To determine whether the LLOQ assessed by MultiQuant analysis corresponded to the automated OpenSWATH FDR-based analysis, we plotted the LLOQ (MultiQuant) and the lowest concentration detected by OpenSWATH for the peptides in groups A and B that span the low-attomole to low-femtomole

**a**

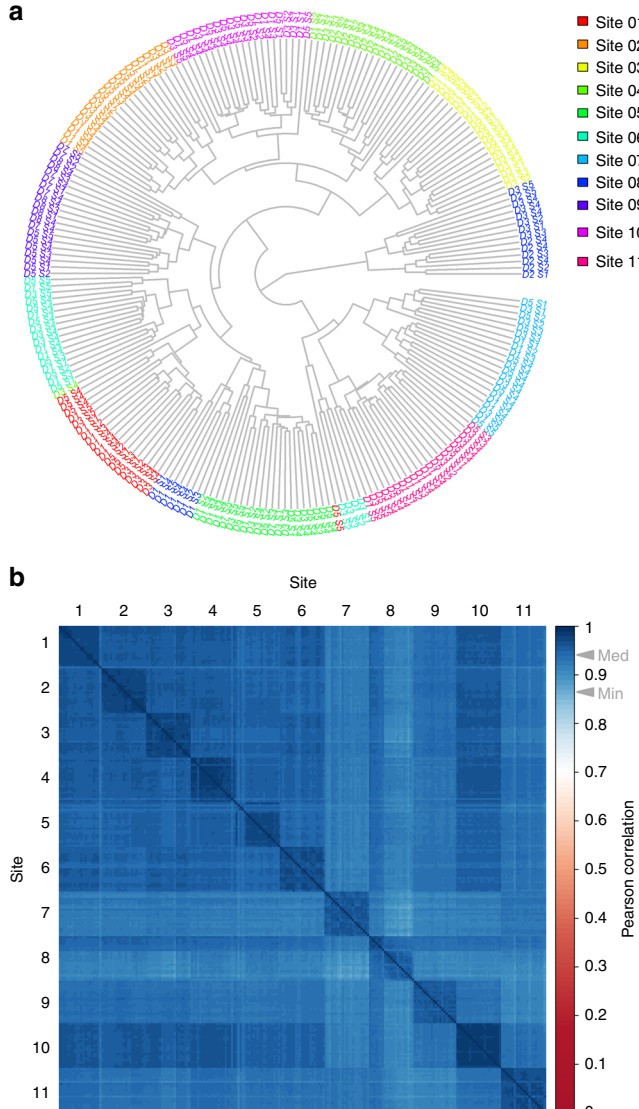

**b**

**Fig. 6** Clustering and correlation of SWATH-MS quantitative proteomes. **a** The dendrogram for the 229 samples from all sites resulting from hierarchical clustering based on the log2 protein abundances generated from the SWATH-MS data is shown. The sites are color coded as per the legend. The "D" and "S" notation refers to the day and sample number respectively (Fig 1a). The samples primarily cluster by site of data acquisition whereas the day of data acquisition with one site is generally not clustered. **b** A correlation matrix showing Pearson coefficients between the 229 samples (all vs. all) is shown. The samples are ordered first by site and then chronologically. The color-scale indicates the magnitude of the Pearson correlation coefficient and the *gray arrowheads* on the color-scale indicate the median and minimum Pearson correlation across all binary comparisons

range (Supplementary Fig. 18). For eight detectable peptides in these groups, six had an LLOQ at the same concentration as the lowest detectable by the FDR-based OpenSWATH analysis and the remaining two peptides had a difference of one 3× dilution step, indicating a good agreement between these methods. We further examined the correspondence in linearity of all SIS peptides as determined by MultiQuant or OpenSWATH and found this to be comparable over the majority of the concentration range, however, OpenSWATH failed to fully integrate very wide chromatographic peaks in the 3–10 pmol range which

resulted in saturation for these concentrations (Supplementary Note 3, Supplementary Fig. 19).

As the SWATH-MS acquisition method also contains an MS1 scan in every cycle, we were able to extract XICs at the MS1 level and determine the LLOQ in MS1 mode using similar criteria for evaluating the individual peptide concentration curves and the LLOQ as was used for the SWATH-MS data (Fig. 5b). Average lines were computed for each mode of quantification and plotted together for easy visualization (Fig. 5c, d). In our data set the LLOQ of peptides using SWATH-MS2 quantification is nearly 1 order of magnitude lower than in MS1. The benefit in this case is explained in terms of selectivity but not absolute signal abundances. While the signal intensity of the precursor in MS1 is typically higher than the fragment ions from the SWATH-MS signal, the MS1 XICs become contaminated with interfering signals as the LLOQ is approached, whereas the SWATH-MS signal generally has less interference at lower analyte concentrations (Supplementary Figs. 20–22, Supplementary Note 4, and Supplementary Data 11 and 12). As with SWATH-MS data, manual inspection of the MS1 data was performed and low concentration peaks not meeting LLOQ requirements were removed. This difference between SWATH-MS and MS1 level sensitivity has also been previously reported[30, 35], although usually with smaller differences between MS1 and SWATH-MS LLOQs that may be explained by the higher complexity of the sample matrix in this study or by the increased number of precursor isolation windows with reduced width compared with previous analyses. Additionally, when compared to the SWATH-MS result, the MS1 data yielded a more divergent detection rate at each concentration across sites, demonstrating that MS1 profiling has a less consistent sensitivity between labs. SWATH-MS demonstrated improved intra-lab reproducibility compared with MS1 with CV values of 8.8 and 13.2%, respectively (Supplementary Fig. 23).

**Global similarity of quantitative protein abundance profiles**. Finally, we elected to examine the global similarity of the normalized quantitative protein abundances determined by SWATH-MS across the different sites of data collection. We performed a hierarchical clustering of the study-wide log2 protein abundance matrix and plotted the resulting dendrogram in Fig. 6a. The data broadly clusters by site of data collection, whereas the day of data collection within one site generally does not cluster. To determine the similarity of the protein abundance profiles more quantitatively, we computed a pairwise Pearson correlation matrix based on the normalized log2 protein abundances of the common proteins from each pair of runs (Fig. 6b). The median Pearson correlation of log2 protein abundances across the entire data set was 0.940. On average, the median Pearson correlation within a given site of data collection was only slightly higher at 0.971 (the range of site medians was 0.948–0.984). The minimum pairwise Pearson correlation between any two of the 229 files across the study was 0.868. From the above analyses, we can conclude that the quantitative similarity within sites of data collection is only marginally higher than between sites of data collection.

### Discussion
The importance of quantitative proteomics in clinical and basic research is expanding rapidly because proteins provide a direct insight into the biochemical state of the cell. To determine the utility of particular proteomic technologies a thorough and objective assessment of their performance is essential. For the widespread application of the technology, robustness, reproducibility, quantitative accuracy, data comprehensiveness and

completeness are critically important performance parameters[62]. Targeted proteomics via SRM is a proven technology receiving high grades with respect to these metrics. The Clinical Proteomic Technologies for Cancer Initiative as part of the Clinical Proteomic Tumor Analysis Consortium (CPTAC) projects[24, 26, 27] have demonstrated that the robust application of SRM across different labs is achievable and an Atlas of SRM assays for the entire human proteome has been published[63]. These results suggest that distributed studies with hundreds to thousands of samples and data integration between labs are becoming feasible. They also generally increased the confidence that smaller and larger scale, comparative proteomic studies are a reality. However, the feasibility of larger scale sample comparisons on protein numbers which exceed that quantifiable by SRM by orders of magnitude has not been demonstrated. SWATH-MS is a technique that has the potential to achieve this ambitious objective. The goal of our study was to characterize the performance of SWATH-MS data acquisition across different laboratories.

The data set analyzed in this study supports a number of conclusions relating to the above stated questions. Firstly, the set of proteins we detected across all sites is very similar and is effectively saturated after a small number of files are analyzed. This indicates that the level of data completeness from a protein quantification perspective is very high, a quality which is desirable in comparative studies. In this study, we have evaluated technical reasons for missing data in relation to measurement variation. Challenges associated with missing data related to biological variation are discussed in Supplementary Note 5.

Notably, the spectral library and peptide query parameters we used to perform the analysis of the SWATH-MS data were previously published[49] and built by a single lab independent of the current study, illustrating the generic applicability of such spectral libraries. Appropriate FDR control was key to achieving this result. Extending the FDR control to the global context (computed over all files in the analysis), in addition to extending the FDR control from the peptide query to the protein level, were critical in the project where large numbers of samples were analyzed using a large number of peptide queries. In a related manuscript we discuss issues relating to FDR control in DIA data in detail[55].

We expect that a DDA-based study could not achieve such a high level of completeness across labs due to stochastic MS2 sampling[7] and such a study is likely to experience difficulty aligning MS1 signals arising from different labs where chromatography will inevitably vary (Supplementary Fig. 4). Importantly, our analysis method did not employ any alignment or propagation of peptide identifications as is commonly used in MS1 quantification from DDA data, however, we anticipate that data completeness might be further improved using a feature alignment strategy recently developed for SWATH-MS[64]. Secondly, the quantitative characteristics in terms of reproducibility, limit of detection, and linear dynamic range were also highly comparable across the data from all sites. Again, with regard to large-scale proteome quantification (i.e., 4000+ proteins) across laboratories in >200 measurements, these findings are unprecedented and have evolved to a level where many of the previously described limitations of data acquisition in MS-based proteomics[62] are being significantly overcome.

In the course of analyzing the data, some interesting characteristics of SWATH-MS data became apparent. For example, one observation relates to the absolute signal response of instruments from various sites, which as expected, was variable. Interestingly, the slope, linearity and dynamic range of the response curves from the SIS peptide dilution series are essentially uniform across sites with only an offset in the intensity dimension differing (Fig. 4c). Further, the number of proteins detected at a given site was only moderately correlated with signal intensity (Supplementary Fig. 8). This suggests that the absolute signal intensity is not the critical metric in determining the data quality, but probably rather the signal-to-noise ratio. These observations have important consequences for normalization of label-free quantitative data and, in our study, facilitated the use of a simple global median normalization based on all of the available peptide signals from the HEK293 background proteome to effectively make the data comparable without the use of internal standards. Here, we highlight an advantage of SWATH-MS data; i.e., as with MS1/DDA-based quantification and, unlike more classical targeted methods such as SRM, there are large numbers of peptides available for global normalization that can be used in sample types where the assumptions underlying this type of normalization are valid[65, 66]. This data set may also be useful for future optimization of certain general data analysis parameters, such as, selection of the most appropriate peptides for protein quantification. In this study, we used a simple method to infer protein abundance[44], however, more advanced methods that take into account which peptides are most robust for quantification ("quantotypic"[67]) across the study could be developed based on our data.

Another comparison that was directly possible in our data set was that of LLOQ in either SWATH-MS or MS1 mode using XIC based analysis within the same data files. As previously reported, we found a clear benefit in sensitivity when extracting quantitative information from SWATH-MS data over MS1 data. This difference was maintained across all sites where the data was acquired, and seems to be generalizable at least with respect to the instrument setup used in this study. It should be stressed that this effect may be somewhat platform dependent, as mass analyzers with higher resolving power for MS1 spectra would facilitate smaller XIC widths, reducing interferences to some degree.

Finally, a further comparison with CPTAC and associated projects focused on targeted proteomics via SRM is of interest as it represents the most advanced work on the robustness and transferability of quantitative proteomics methods to date[24, 26, 27]. CPTAC has also published inter-lab studies focused on DDA analysis. However, these have primarily focused on the repeatability of peptide/protein identifications or the establishment of quality control metrics[7, 10], or on higher level similarity of differential expression analysis when different instruments and quantitative approaches were applied[66], but have not addressed specific comparisons of quantitative metrics such as CV, LLOQ, linearity, or dynamic range. Our study is conceptually related to what was achieved by the CPTAC SRM studies although there are also some major differences. Firstly, the scope of the CPTAC SRM studies was different and included variables such as sample preparation, system suitability, and instrumentation from different vendors. In the case of our study, the decision to include only a single instrument type and model was primarily to limit the number of experimental parameters varied and, secondly, because at the outset of the project (September 2013) the adoption of SWATH-type DIA analysis on other platforms was limited. As such, in our study, the main variable tested was the site of data acquisition to assess inter-laboratory SWATH data quality and reproducibility. As we did not evaluate the variance in sample preparation between sites we cannot make any conclusions on this topic. However, we would suggest that the conclusions in the CPTAC analysis are generalizable; i.e., that if samples are prepared at different sites a significant batch effect can be expected. As such, viable options for future distributed studies would be to prepare the samples at central facility or to invest significantly in standardization of sample preparation in combination with the application of more advanced methods for normalization and removal of batch effects. Another significant design difference is

that CPTAC SRM studies were focused on achieving essentially clinical-grade assays[68] for relatively discrete sets of targets. Our focus was on quantifying large numbers of proteins in a workflow that might be used either in a discovery mode for hypothesis generation, or in a verification mode to test large numbers of protein analytes in large cohorts. Lastly, as CPTAC has been focused on a relatively discrete set of targets it was possible to include isotope-labeled standards, which helped to determine absolute concentrations and to control matrix interference effects, whereas our study focused on label-free analysis. With these differences stated, we can suggest that our studies lead to a conceptually similar conclusion, albeit with different scopes. That is, using either targeted MS (i.e., SRM) to study discrete panels of proteins with highly validated assays or using DIA (i.e., SWATH-MS) to study large numbers of proteins in exploratory/verification analyses, we can quantify proteins in a robust and complete manner.

This study has demonstrated for the first time that large-scale quantification of several thousand proteins from centrally prepared samples is feasible with reproducible and comparable data generated across multiple labs. The result of our study, focused on assessing variation in data acquisition, is paralleled by concurrent improvements in the robustness of data analysis tools[39], methods for error rate control[55], and sample preparation techniques[43]. While further work needs to be done in several areas, such as large-scale sample preparation, long-term instrument robustness, and batch effect normalization during data analysis, these studies collectively advance the reproducibility and transparency of SWATH-MS. As comparative quantitative analysis of a large number of proteomes becomes accessible[42, 46, 69], we can expect to see research applications where the analysis of large numbers of samples is a prerequisite. For example, analyses of clinical material from large patient cohorts[42] (e.g., biomarkers, personalized medicine), association of protein abundances to genomic features using genetic reference collections or wild-type populations[46] (e.g., quantitative trail locus or genome wide association studies), or large-scale perturbation screens using in vitro model systems (e.g., drug screens) are now feasible. More broadly, the data presented here demonstrate a significant advance in the robustness of large-scale data acquisition in quantitative proteomics, and we expect the results from this study to increase confidence in SWATH-MS as a reproducible quantification method in life science research.

## Methods

**Generation and distribution of a benchmarking sample.** HEK293 cells (ATCC—low passage cells—not verified or mycoplasma tested) were cultured in DMEM (10% FCS, 50 µg ml$^{-1}$ penicillin, 50 µg ml$^{-1}$ streptomycin). HEK293 cells were selected as they are a common cell line used in molecular biology research with many published orthogonal data sets. Cell pellets were lysed on ice by using a lysis buffer containing 8 M urea (EuroBio), 40 mM Tris-base (Sigma-Aldrich), 10 mM DTT (AppliChem), and complete protease inhibitor cocktail (Roche). The mixture was sonicated at 4 °C for 5 min using a VialTweeter device (Hielscher-Ultrasound Technology) at the highest setting and centrifuged at 21,130×$g$, 4 °C for 1 h to remove the insoluble material. The supernatant protein mixtures were transferred and the protein amount was determined with a Bradford assay (Bio-Rad). Then five volumes of precooled precipitation solution containing 50% acetone, 50% ethanol, and 0.1% acetic acid were added to the protein mixture and kept at −20 °C overnight. The mixture was centrifuged at 20,400×$g$ for 40 min. The pellets were further washed with 100% acetone and 70% ethanol with centrifugation at 20,400×$g$ for 40 min. Aliquots of 2 mg protein mixtures were reduced by 5 mM tris(carboxyethyl) phosphine (Sigma-Aldrich) and alkylated by 30 mM iodoacetamide (Sigma-Aldrich). The samples were then digested with sequencing-grade porcine trypsin (Promega) at a protease/protein ratio of 1:50 overnight at 37 °C in 100 mM NH$_4$HCO$_3$ (ref. [70]). Digests were combined together and purified with Sep-Pak C18 Vac Cartridge (Waters). The peptide amount was determined by using Nanodrop ND-1000 (Thermo Scientific). An aliquot of retention time calibration peptides from an iRT-Kit (Biognosys) was spiked into the sample at a ratio of 1:20 or 1:25 (v/v) to correct relative retention times between acquisitions[71].

Thirty heavy labeled synthetic peptides that were previously used in an SRM study focused on limits of detection in mammalian cells[48] were selected. As such, these peptides are expected to perform well in LC–MS analysis. The MS response for each peptide was measured. The peptides were ranked by MS response and assigned to five groups (A–E) to ensure there was a range of responses across in each group. These peptides groups were diluted into the matrix described above across a concentration range to create the five different samples to be analyzed (Fig 1a, Supplementary Tables 1 and 2). Finally, samples were shipped on dry ice to the 11 sites.

**SWATH-MS measurements.** Peptide mixtures were separated using reversed phase nanoLC using either a nanoLC Ultra system or a nanoLC 425 system (SCIEX). Most sites (9 of 11) used a cHiPLC system (SCIEX) operated in serial column mode (for detailed acquisition information please see SOP in Supplementary Protocol 1), fitted with two cHiPLC columns (75 µm × 15 cm ChromXP C18-CL, 3 µm, 300 Å) to give a total column bed length of 30 cm (Site configuration details in Supplementary Table 13). Two sites used PicoFrit emitter (New Objective) packed to 30 cm with Magic C18 AQ 3 µm 200 Å stationary phase. Peptide samples (2 µL injection) were first loaded on the first cHiPLC column and washed for 30 min at 0.5 µl min$^{-1}$ using mobile phase A (2% acetonitrile in 0.1% formic acid). Then, elution gradients of ~5–30% of mobile phase B (98% acetonitrile in 0.1% formic acid) in 120 min were used to elute peptides off the first column and through the second cHiPLC column. Both columns were maintained at 35 °C for retention time stability. Similar separations were performed across all sites. Gradients were allowed to minimally vary from site to site to obtain similar peptide separations (see Supplementary Table 14 for gradient information).

Eluent from the column was introduced to the MS system using the NanoSpray Source into a TripleTOF 5600 system with Analyst Software TF 1.6 (SCIEX) and the variable window acquisition beta patch. The SWATH-MS acquisition methods were built using the SWATH-MS Acquisition method editor and a pre-defined variable window width strategy using 64 windows (Supplementary Table 15). The Q1 mass range interrogated was 400–1200 $m/z$, and MS2 spectra were collected from 100 to 1500 $m/z$ with an accumulation time of 45 ms per variable width SWATH window. A TOF MS scan (250 ms, 400–1250 $m/z$) was acquired in every cycle for a total cycle time of ~3.2 s. Nominal resolving power for MS1 and SWATH-MS2 scans were 30,000 and 15,000 respectively. The collision energy curve was controlled across all instruments (CE = 0.0625 * $m/z$ − 3) and the collision energy spread was defined in the variable window table (Supplementary Table 15). The acquisition order is outlined in the Supplementary Table 16. SWATH-MS data files (2 out of 231) were excluded by the local operators if there was an obvious acquisition error.

**Pilot phase quality control assessment.** SWATH-MS acquisition data from the pilot study phase were processed using the SWATH® Acquisition MicroApp 2.0 in PeakView Software 2.2. A previously published proteome library containing mass spectrometric coordinates for 10,000+ human proteins[49] was used for data processing. iRT standard peptides (Biognosys) were included in the library for automatic retention time calibration of each different sample set with the ion library retention times. Peak group detections were filtered at a 1% global FDR and metrics were compared using Excel (this corresponds to data in Supplementary Fig. 1 only).

**Automated analysis of SWATH-MS data.** The SWATH-MS data analysis was performed using OpenSWATH (OpenMS v2.0) essentially as described[35] except that the improved single executable OpenSwathWorkflow was used instead of the multi-step workflow to perform peak-picking and feature detection and the following parameters were changed: $m/z$ extraction window = 75 ppm, RT extraction window = 900 s. The spectral library used as input for peptide queries in the OpenSWATH analysis was a previously published proteome library containing mass spectrometric coordinates for 10,000+ human proteins built by combining several hundred DDA analyses of various human cell and tissues types[49].

Semi-supervised learning to optimally combine OpenSWATH peptide query scores into a single discriminant score, and $q$-value[50] estimation to facilitate FDR control, were performed using an extended version of PyProphet[72] (PyProphet-cli v0.19—https://github.com/PyProphet). PyProphet was run both using the experiment-wide context (local–global option in PyProphet—$q$-values are generated for every peptide query and protein in every sample) and the global context (global–global option—only one $q$-value for every peptide query and protein representing the highest scoring instance over the whole experiment), with a fixed $\lambda$ of 0.4. The set of peptide peak groups used for learning the score weights of OpenSWATH sub-scores to produce a single discriminant score were sampled with a ratio ≈1/(no. of samples) in the analysis (for aggregated analysis of all sites 0.005, and for analysis of individual sites 0.05). The sets of peak groups detected at 1% FDR and proteins detected at 1% FDR in the global context were used as a filter to restrict the set of peak groups and proteins in the experiment-wide context. The filtered table from the experiment-wide context was then filtered at 1% FDR at the peptide query level. A protein was considered as detected in a given sample if it passed these consecutive filters (see Supplementary Note 2 for further discussion on FDR control). The repeatability[7] was defined as the intersect divided by the union between the peptide or proteins detected from two data files computed pairwise within the site of data collection or across the entire data set.

Normalization was achieved by equalizing medians at the peak group level. The normalization coefficients derived from the peak groups in HEK293 matrix were also used to normalize the peak areas determined by MultiQuant analysis (below) of the SIS peptides. Protein abundances were inferred by summing the top five most intense fragment ion peak areas from the top three most intense peak groups using the aLFQ software[57] (v1.33). Where <3 peak groups were detected, the available peak groups were summed. Coefficients of variation (% CV) were computed as 100*standard deviation/mean. Hierarchical clustering was performed using the dist and hclust functions in R (v3.2.2) using log2 transformed protein abundances and visualized using the R package ape (v3.3). Pearson correlation coefficients were computed using the R package Hmisc (v3.17) and visualized using the R package corrplot (v0.73).

**Analysis of SWATH-MS data for 30 SIS peptides**. The SWATH Acquisition data obtained from all sites was processed using MultiQuant Software 3.0. The same quantification method (Supplementary Table 17) was used across all sites and consisted of three to four fragment ion XICs extracted and summed together to produce a peptide area. Spectral peak widths for XIC generation were 0.05 for MS2 and 0.02 for MS. Peak integration was done using the MQ4 algorithm. The curve for each peptide was evaluated and the LLOQ was determined in accordance with bioanalytical standards[49] (<20% CV, $S/N > 20$, 80–120% accuracy; linear fit with $1/x$ weighting). A number of analytical aspects were evaluated, including the reproducibility of the peptide peak areas, the LLOQ for each peptide, the signal/noise ratios using the relative noise approach in the MultiQuant Software, and the reproducibility and accuracy of the concentration.

**Data availability**. The mass spectrometry proteomics data has been deposited to the ProteomeXchange Consortium (http://proteomecentral.proteomexchange.org) via the PRIDE partner repository[73] with the data set identifier PXD004886. The data that support the findings of this study are available from the corresponding author upon request.

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

## Acknowledgements

We thank Alex Ebhardt for providing the SIS peptides for this study; Eric Deutsch for facilitating FTP data exchange; Isabell Bludau for discussions on FDR control; Uwe Schmitt for development of the PyProphet extension; Hannes Röst for discussions on normalization and data analysis; Emanual Schmid for assistance with data management. We thank Asa Wahlander and Bernd Roschitzki from the Functional Genomics Center Zurich (FGCZ) for instrument maintenance and support with the MS measurements. A.-C.G. is the Canada Research Chair in Functional Proteomics and the Lea Reichmann Chair in Cancer Proteomics. We acknowledge funding from the Government of Canada through Genome Canada and Ontario Genomics (OGI-088, OGI-097) and Canadian Institutes of Health Research (FDN-143301) to A.-C.G.; the National Cancer Institute Clinical Proteomics Tumor Analysis Consortium (CPTAC) grant U24CA160036 to D.W.C. and H.Z.; Chinese National Basic Research Programs (2014CBA02002, 2014CBA02005). N.S. is supported by funding from the European Union's Seventh Framework Program HEALTH-F4-2013-602156. We acknowledge support from the NIH shared instrumentation grant for the TripleTOF system at the Buck Institute (1S10 OD016281, B.W.G.). M.P.M. acknowledges support from the Australian Government's National Collaborative Research Infrastructure Scheme. This work was funded in part by National Institutes of Health Grant RC2 HG005805 from the National Human Genome Research Institute (NHGRI) through the American Recovery and Reinvestment Act and Grants from the National Institute of General Medical Sciences (NIGMS) grants R01GM087221, S10RR027584 and 2P50GM076547 to the Center for Systems Biology, the National Science Foundation grant MCB-1330912, AMED-CREST from Japan Agency for Medical Research and Development, and the Funding Program for Next Generation World-Leading Researchers by the Cabinet Office to M.H.-K. and S.O. B.C.C. was supported by a Swiss National Science Foundation Ambizione grant (PZ00P3_161435). R.A. was supported by ERC Proteomics v3.0 (AdG-233226 Proteomics v.3.0) and AdG-670821 Proteomics 4D), the PhosphonetX project of SystemsX.ch and the Swiss National Science Foundation (SNSF) grant number: 31003A_166435.

## Author contributions

B.C.C., C.L.H. and Y.L. prepared the samples, analyzed the data, and wrote the manuscript. B.S. contributed to protocol preparation and manuscript writing. G.R. assisted with data analysis. B.C.C., C.L.H., Y.L., B.S., S.L.B., D.W.C., B.W.G., A.-C.G., J.M.H., M.H.-K., G.H., C.K., B.L., L.L., S.L., M.P.M., R.L.M., S.O., R.S., N.S., S.N.T., S.-C.T. and H.Z. acquired the data and contributed to manuscript writing. B.C.C., C.L.H., Y.L. and R. A. designed and directed the study.

## Additional information

**Competing interests:** C.H. is an employee of SCIEX, which operates in the field covered by the article. R.A. holds shares of Biognosys AG which operates in the field covered by the article. The remaining authors declare no competing financial interests.

