## [Peer Review File · Nature Communications]

Editorial Note: This manuscript has been previously reviewed at another journal that is not operating a transparent peer review scheme. This document only contains reviewer comments and rebuttal letters for versions considered at Nature Communications. Mentions of prior referee reports have been redacted.

PEER REVIEW FILE

Reviewers' Comments:

Reviewer #1 (Remarks to the Author):

Collins et al. present a well-conceived and well executed intra-laboratory study investigating the performance of a data-independent acquisition method (SWATH-MS) on a single type of instrument (TripleTOF 5600).

The study shows quite elaborately that SWATH MS performs (after some optimization) highly consistent across multiple laboratories.

The authors compare the data from a benchmarking sample which was acquired by 11 different laboratories worldwide. The analyses are very comprehensive and show:

- a) proteins are consistently detected
- b) FDR control is required when combining data from high numbers of runs
- c) Proteins are reproducibly quantified
- d) Data show a good linearity and 4.5 orders of magnitude dynamic range
- e) Sensitivity of SWATH-MS is higher in MS2 than in MS1
- f) Protein abundance profiles are similar between the different sites

The study shows clearly that SWATH-MS enables a significant advance in the robustness of label-free quantitative proteomic analysis. This study is a significant advance for the field of quantitative proteomics.

Materials and Methods are described in high detail and enable other researchers to reproduce the analyses and results. The study design is very well presented and all raw data are provided. The approach pursued by the authors is very comprehensive and great attention has been paid to methodological detail. Data are of exceptionally high quality and results are concisely presented.

The authors have significantly improved their manuscript in response to the various reviewer

comments.

Reviewer #2 (Remarks to the Author):

The authors have addressed all of my concerns from the previous review, and in my opinion the manuscript is acceptable for publication as is. I appreciate the careful consideration the authors put into addressing the concerns of myself and the other reviewers.

I was curious about how the spike-in data would look when comparing the response curves of individual peptides, rather than by site. The authors mention in the response to previous review that this would be of interest in a "targeted proteomics" study, and I agree. I attached the plots I generated below (output.pdf). I would encourage the authors to include similar plots in the manuscript supplement. I think that they are a useful way to give readers a more detailed understanding of how widely the quantification of an individual peptide could vary between sites. I leave this decision up to the authors and reiterate that including plots like these are not a condition for acceptance of this manuscript.

I also recommend that the authors update the legend for Figure 4 to specify how the data were averaged (I think it's median).

Reviewer #3 (Remarks to the Author):

The reviewers comments were all satisfyingly addressed and the manuscript is ready for publication. The study provides a complete and in depth analysis of the results and provides useful hints how to conduct further large scale studies. Also, the data set is made publicly available and will provide a very valuable resource for futur research. The scope should be made more explicit in the abstract of the paper. Otherwise I have no further comments.

Comments:

1) In your "General comments to reviewers regarding scope" you state "In our paper we specifically address the issue of data acquisition.". This important point should be stated in the Abstract as well. It should be stated that the sample preparation and data analysis were conducted by a central lab, and data acquisition using SWATH was distributed.

2) Line 54: "we have shown" -> "we show"